# Factors Associated with Overweight and Obesity in Adults from Rio Branco, Acre in the Western Brazilian Amazon

**DOI:** 10.3390/nu14051079

**Published:** 2022-03-04

**Authors:** Yara de Moura Magalhães Lima, Fernanda Andrade Martins, Alanderson Alves Ramalho

**Affiliations:** 1Postgraduate Program in Public Health, Federal University of Acre, Rio Branco 69920-900, AC, Brazil; yara.moura@sou.ufac.br; 2Center for Health Sciences and Sports, Federal University of Acre, Rio Branco 69920-900, AC, Brazil; fernanda.martins@ufac.br

**Keywords:** obesity, overweight, nutrition surveys, health surveys

## Abstract

This study aimed to assess factors associated with overweight and obesity in adults from Rio Branco, Acre, in the western Brazilian Amazon. This is a cross-sectional, population-based study conducted in Rio Branco, which used data on individuals aged 18 years or older collected by the 2019 National Health Survey. Software R version 4.0.5 was used to estimate the prevalence of overweight and obesity, prevalence ratios, and 95% confidence intervals. Multiple analysis was performed by Poisson’s regression with robust variance and hierarchical selection of variables. This study included 1217 adults. The prevalence of overweight was 58.2% (95%CI: 54.7–61.6) and of obesity, 20.1% (95%CI: 17.2–23.4). The factors associated with overweight were arterial hypertension (AdjPR: 1.45; 95%CI: 1.31–1.61), physical inactivity (AdjPR: 1.19; 95%CI: 1.04–1.36), age group (25–39 years, AdjPR: 1.49; 95%CI: 1.10–2.00; 40–59 years, AdjPR: 1.69; 95%CI: 1.28–2.23; 60 years or older, AdjPR: 1.37; 95%CI: 1.01–1.87); and smoking (AdjPR: 0.62; 95%CI: 0.41–0.93). The factors associated with obesity were arterial hypertension (AdjPR: 1.80; 95%CI: 1.41–2.30) and diabetes mellitus (AdjPR: 1.52; 95%CI: 1.08–2.13). Smoking and female sex remained in the hierarchical model for obesity, even without statistical significance. Despite intervention guidelines for these chronic diseases, there is a need for the public recognition of overweight and obesity and their possible associated factors in the Amazon and other regions with similar socioeconomic and demographic characteristics.

## 1. Introduction

Chronic non-communicable diseases (NCDs) have become one of the leading causes of death and disability in the world. In 2017, overweight and obesity caused 34.1 million deaths and 1.2 billion disabilities, values significant for both sexes [1]. These and other consequences highlight overweight and obesity as diseases with considerable clinical and economic impact. In 2014, the global economic impact attributable to obesity reached approximately USD 2 trillion per year and 2.8% of the global GDP [2].

The progressive growth in the prevalence of overweight and obesity has been evidenced worldwide. According to the World Health Organization, the prevalence of overweight in 2016 affected more than 1.9 billion adults (39%), of which, 650 million (13%) showed some degree of obesity. Once associated with high-income countries, this reality now affects most nations [3,4,5].

In Brazil, according to the Surveillance System of Risk and Protective Factors for Chronic Diseases by Telephone Survey (Vigitel), the prevalence of obesity in individuals aged 18 years or older increased from 11.8% in 2006 to 20.3% in 2019, a 72% increase. Moreover, the number of people with obesity has grown in all regions of the country. The trend may be associated with urbanization, increased consumption of ultra-processed foods, reduced consumption of natural foods, and increased physical inactivity. A nutritional transition involves behavioral, geographic, cultural, and socioeconomic factors. Thus, regional differences must be considered and analyzed individually [6,7].

Located in the Brazilian Amazon, Rio Branco, the capital of Acre State, shows one of the highest upward trends of overweight and obesity in the region. From 2006 to 2020, the annual percentage change of overweight was 5.2% (95%CI: 1.4–9.1) until 2010 and 1.3% from 2010 to 2020. This upward trend was also observed in the prevalence of obesity: 7.7% (95%CI: 4.5–10.9) until 2012 and 1.7% (95%CI: −0.2–3.7) from 2012 to 2020 [8]. Thus, this study aimed to assess factors associated with overweight and obesity in adults from Rio Branco, Acre in the western Brazilian Amazon.

## 2. Materials and Methods

This is a population-based, cross-sectional study which uses the database of the National Survey on Health (PNS), conducted by the Brazilian Institute of Geography and Statistics (IBGE) in partnership with the Ministry of Health. The population analyzed were individuals aged 18 years or older who participated in PNS data collection and resided in Rio Branco, Acre. The municipality is the capital of the Acre State, located in Northern Brazil, and has 413,418 inhabitants, concentrating 46.2% of the total population of the state (Figure 1) [9].

The PNS is a household survey which employs a cluster sampling plan in three selection stages: (I) primary sampling units (census tracts–UPA), (II) secondary units (households/UPA), and (III) tertiary units (resident aged 15 years or older who answered individual questionnaires). Thus, the information collected included questions on the household, its residents, and the adult selected by sample draw. Initially, contact was made with a resident of the selected household, and the study were presented. With the consent to participate, the household questionnaire was applied to the informant and the individual questionnaire to the selected resident. For this study, respondents aged 18 years or older were selected, excluding pregnant women and women with suspected pregnancy. As it is a complex sample, estimates were considered by the weight on eligible adult residents, corresponding to the capital and state analyzed, enabling a representative sample of the population of Rio Branco, Acre in 2019 [10].

In this study, our outcomes of interest were overweight and obesity, evaluated according to weight and height measurements collected during individual interviews. Diagnoses were made via the body mass index (BMI), estimated from the selected PNS participants’ weight (in kilos) divided by the square of their height (in meters). A BMI ≥ 25 kg/m^2^ was considered overweight and BMI ≥ 30 kg/m^2^ was considered obesity. Independent variables were also investigated: sociodemographic characteristics (age group, sex, and ethnicity), economic characteristics (per capita income and possession of health insurance), lifestyles (regular fruit and vegetable consumption, regular soft drink and/or refreshment consumption, regular ultra-processed food consumption, smoking, and alcohol abuse), and health determinants (physical inactivity, self-reported poor health, diabetes mellitus, hypertension, and dyslipidemia) [10].

Age groups were considered by residents’ ages in years. For analysis, they were stratified into 18–24, 25–39, 40–59, and 60+. Sex was collected via the binary alternatives of the instrument, male or female, since the database used does not work with genders. Ethnicity was categorized by the alternatives: white, black, yellow, brown, and Indigenous. For analysis, the white ethnicity was maintained, whereas remaining alternatives were recategorized into “other” in view of the difficulties of self-perception and phenotypic self-declaration of skin color, especially in multiracial countries such as Brazil [11,12]. Economic variables, such as per capita income, were estimated by the total family income divided by the number of household residents and expressed in minimum wages. In 2019, the minimum wage in Brazil was BRL 998.00, equivalent to USD 187.95 in 2022. The variable health insurance was reported by interviewees by the alternatives: yes or no [10].

Regular fruit and vegetable consumption was considered when both foods were consumed on five or more days of the week, using as the minimum consumption, a fruit (including juice) and a type of salad or cooked vegetable per day. Regular soft drink and/or refreshment consumption, categorized as yes or no, was considered for individuals who reported consuming soft drinks or refreshments/artificial juices on five or more days a week. Regular ultra-processed food consumption, categorized as yes or no, was considered when at least five ultra-processed foods were consumed in the day before the research. To assess this, questions focused on the consumption of possible ultra-processed foods, such as artificially flavored sweetened beverages, snacks or industrialized cookies, industrialized desserts, sausages in general, and industrialized bread, pasta, and sauces. Smoking, categorized as yes or no, was considered if present, regardless of type, quantity, frequency, and duration. Alcohol abuse, categorized as yes or no, was considered when alcohol had been excessively consumed in the last 30 days prior to research. PNS considered abuse, for both men and women, the consumption of five or more doses of alcoholic beverages on a single occasion [10].

Physical inactivity was assessed through questions related to frequency (days per week) and duration (time per day) in different domains, such as free time or leisure, occupational activity, during commuting, and within the scope of domestic activities. This dichotomous variable was considered for the individual who practiced physical activity for less than 20 min a day, or who did not practice any activity in the last three months prior to research. Self-reported poor health, categorized as yes or no, was considered for individuals who negatively evaluated their own health status (i.e., as poor or very bad). Diabetes mellitus, hypertension, and dyslipidemia, categorized as yes or no, were considered if participants reported previous related medical diagnoses, except for women who received diagnoses during pregnancy [10].

The statistical analysis of this complex sampling was performed via software R, version 4.0.5. To assess independent variables as possible factors associated with overweight and obesity in the population studied, a bivariate analysis was initially performed to estimate prevalence ratios (PR) and their respective 95% confidence intervals (95%CI). Next, the Poisson regression with robust variance was used, and variables showing statistical significances of up to 20% (*p* ≤ 0.20) were selected for insertion in the multiple model. In our multiple analysis, the hierarchical entry of independent variables was used following distal, intermediate, and proximal orders [13], according to the hierarchical model shown in Figure 2. Variables that improved the adjustment were retained in the final model and remained statistically significant (*p* < 0.05).

Ethical appreciation was waived for this research since it employed public-use data, unrestrictedly made available, without nominal identifications, by the Brazilian Institute of Geography and Statistics (IBGE) under the Resolution of the National Health Council, CNS 466/12, providing for research with human beings in Brazil [10].

## 3. Results

This study included 1217 individuals, aged 18 years or older, who were interviewed by the PNS household survey in 2019, in the city of Rio Branco, the capital of Acre State, Brazil. 

In our sample, 35.3% of respondents were aged from 40 to 59 years old, 53.0% were women, 79.6% declared an ethnicity other than white, 27.7% received up to half a minimum wage, and 82.4% had no health insurance (Table 1).

The prevalence of overweight was 58.2% (95%CI: 54.7–61.6) and of obesity, 20.1% (95%CI: 17.2–23.4).

### 3.1. Prevalence of Overweight and Obesity according to Sociodemographic and Economic Characteristics

Table 1 shows the distribution of the prevalence of overweight and obesity according to sociodemographic and economic variables. The prevalence of overweight was significantly higher among those aged from 40 to 59 years (69.2%; PR: 1.93; 95%CI: 1.54–2.41), 60 years or above (62.3%; PR: 1.74; 95%CI: 1.34–2.26) and from 25 to 39 years (56.6%; PR: 1.58; 95%CI: 1.21–2.05) but similar across sexes. The prevalence of obesity was higher in individuals aged 40 to 59 years (27.5%; PR: 2.43; 95%CI: 1.41–4.19), and women (23.0%; PR: 1.37; 95%CI: 1.04–1.81).

### 3.2. Prevalence of Overweight and Obesity according to Lifestyle and Health

Of the 1217 adults interviewed, 10.8% regularly consumed fruits and vegetables; 2.9%, soft drinks and/or refreshments at least five days a week; and 12.8%, five or more ultra-processed food groups. In our sample, 10.6% smoked, 15.5% abused alcohol, 5.4% self-reported poor health, and 13.9% were physically inactive. Regarding diagnoses of chronic non-communicable diseases, 6.2% had diabetes mellitus; 20.7%, hypertension; and 11.7%, dyslipidemia (Table 2).

Table 2 shows the prevalence of overweight and obesity according to lifestyle and health variables. The prevalence of overweight was lower in individuals who smoked (41.5%; PR: 0.69; 95%CI: 0.49–0.97) and higher in those with comorbidities, such as diabetes mellitus (79.3%; PR: 1.33; 95%CI: 1.16–1.54), hypertension (82.7%; PR: 1.59; 95%CI: 1.43–1.72), and dyslipidemia (72.1%; PR: 1.19; 95%CI: 1.03–1.37), as well as in physically inactive individuals (69.7%; PR: 1.26; 95%CI: 1.08–1.47).

The prevalence of obesity was higher in individuals who regularly consumed fruits and vegetables (27.1%; PR: 1.41; 95%CI: 1.04–1.92), and those with diabetes mellitus (39.3%; PR: 1.96; 95%CI: 1.38–2.77) and hypertension (34.3%; PR: 2.04; 95%CI: 1.61–2.56). However, it was significantly lower in interviewees who smoked (10.3%; PR: 0.48; 95%CI: 0.25–0.94). 

### 3.3. Factors Associated with Overweight

Table 3 shows Poisson regression models referring to adjusted prevalence ratios (AdjPR) of overweight in adults in Rio Branco, Acre, according to sociodemographic, economic, lifestyle, and health variables. In the final model, the factors directly associated with overweight were arterial hypertension (AdjPR: 1.45; 95%CI: 1.31–1.61), physical inactivity (AdjPR: 1.19; 95%CI: 1.04–1.36), and age groups 40 to 59 years (AdjPR: 1.69; 95%CI: 1.28–2.23), 25 to 39 years (AdjPR: 1.49; 95%CI: 1.10–2.00), and 60 years or older (AdjPR: 1.37; 95%CI: 1.01–1.87). Smoking was inversely associated with overweight (AdjPR: 0.62; 95%CI: 0.41–0.93). 

### 3.4. Factors Associated with Obesity

Table 4 shows Poisson regression models referring to adjusted prevalence ratios (AdjPR) of obesity in adults in Rio Branco, Acre, according to sociodemographic, economic, lifestyle, and health variables. In the final model of hierarchical multiple analysis, the factors associated with obesity were arterial hypertension (AdjPR: 1.80; 95%CI: 1.41–2.30) and diabetes mellitus (AdjPR: 1.52; 95%CI: 1.08–2.13). The female sex (AdjPR: 1.19; 95%CI: 0.90–1.57) and smoking (PR: 0.69; 95%CI: 0.37–1.28) remained in our final model but lost statistical significance when we added dyslipidemia, diabetes mellitus, and arterial hypertension.

## 4. Discussion

This study found a high prevalence of overweight and obesity in adults from Rio Branco in 2019. The factors associated with overweight were hypertension, physical inactivity, age group, and non-smoking. The factors associated with obesity were arterial hypertension and diabetes mellitus. The female sex and smoking remained in the hierarchical model for obesity but without statistical significance.

According to Vigitel 2019, the frequency of overweight adults in Brazilian capitals ranged from 49.1% to 60.9%, and that of obese adults, from 15.4% to 23.4%. In the same survey, Rio Branco is among the Brazilian capitals with the highest frequencies of overweight (55.6%) and obesity (23.3%) [6]. These frequencies, estimated by the telephone survey in Rio Branco, are similar to those observed in our population study.

A study which analyzed the overweight and obesity trend in adults in Rio Branco in the last 15 years found a significant upward trend for both sexes. The prevalence of obesity showed an annual percentage increase of 4.2% in males and 4.0% in females [8]. That study suggests that one of the hypotheses for this significant increase in overweight and obesity may be associated with the intense economic, social, and demographic changes that occurred in Rio Branco, such as population growth and aging, expansion and construction of new highways, the inauguration of the first shopping mall in the state, and the first national and international fast food franchises in the city [8].

The factors associated with overweight and obesity found in our study have some similarities to those of other Brazilian capitals. However, we should consider the differences in the magnitude of the associations. 

A cross-sectional study using 2019 Vigitel data on Brazilian capitals observed significant sex differences in overweight prevalence and its associated factors. For men, the factors associated with overweight were ages between 35 and 44 years (PR: 1.87; 95%CI: 1.65–2.12), hypertension (PR: 1.24; 95%CI: 1.17–1.31), and diabetes mellitus (PR: 1.09; 95%CI: 1.01–1.19). For women, the factors were ages between 45 and 54 years (PR: 2.03; 95%CI: 1.77–2.32), hypertension (PR: 1.21; 95%CI: 1.15–1.27), and diabetes mellitus (PR: 1.09; 95%CI: 1.03–1.15). The lowest prevalence of overweight was associated with smoking women (PR: 0.84; 95%CI: 0.76–0.94) and men (PR: 0.86; 95%CI: 0.78–0.96) [14].

Another study using secondary data from the 2013 PNS found that the factors associated with the prevalence of obesity in the Brazilian adult population were the age group from 40 to 59 years for both sexes, hypertension diagnosis in men (OR: 2.84; 95%CI: 2.48–3.25) and women (OR:2.40; 95%CI: 2.19–2.64), and diabetes in men (OR: 2.36; 95%CI: 1.91–2.92) and women (OR: 1.83; 95%CI: 1.58–2.11) [15].

Associations between sex, age, overweight, and obesity are well elucidated in the literature [14,15,16,17,18,19]. In our study, the age group from 40 to 59 years showed the highest prevalence of overweight, followed by the age group from 25 to 39 years. Prevalence regressed for those aged 60 years or older. According to data from the U.S. 2017–2018 National Health and Nutrition Examination Survey conducted with adults, the prevalence of severe obesity was higher among adults aged 40 to 59 years (44.8%), regressing in those aged 60 years or older (42.8%) [16]. A systematic review of 183 countries for the 1980–2013 period found that all ages gained weight. However, the age group from 20 to 40 showed a faster weight gain. Physiological and metabolic alterations resulting from aging may influence greater body weight gain, but the effects of morbidities or higher rates of chronic diseases at older ages may cause weight loss [17].

In our study, the prevalence of obesity was higher in women, corroborating other studies [6,14,15,18,19]. Studies suggest that the higher prevalence of obesity in women may be associated with the higher proportion of older women in samples since physiological changes may reflect weight gain [14,19].

In our study analyzing the factors associated with lifestyles, physical inactivity showed a positive association with overweight. A systematic review of 26 studies corroborates these findings of which twenty (77%) associated physical activity with lower incidences of obesity, four (15%) with physical inactivity with higher incidences of obesity, and two (8%) showed no associations [20]. A population study in the United Kingdom suggests that adults who develop overweight tend to show low levels of physical activity, longer exposure to television, and sleep disorders, emphasizing how modifiable behavioral factors contribute to preventing or promoting chronic diseases [21].

Smoking was negatively associated with overweight in Rio Branco. Cigarettes contain approximately 10 to 15 mg of nicotine of which about 10% of the nicotine in each cigarette is absorbed by the systemic circulation [22]. In addition to enabling psychoactive effects, this molecule also reduces appetite and increases metabolic rate, leading to reduced body weight. Though nicotine-induced mechanisms lead to lack of appetite and magnitudes vary according to age, there is no evidence that smoking can help control appetite [23,24].

As in our study, others also show an association between overweight and other chronic non-communicable diseases, such as hypertension and diabetes mellitus [14,15,25,26]. In the United Kingdom, a study with 367,703 Biobank participants found that the highest BMI was associated with eight to fourteen cardiovascular conditions, including hypertension (OR: 1.10; 95%CI: 1.07–1.12), peripheral arterial disease (OR: 1.07; 95%CI: 1.03–1.12), and coronary artery disease (OR: 1.07; 95%CI: 1.04–1.10) [26]. One possible cause is in the association of fat mass index, predominant in overweight and obesity, with greater left ventricular hypertrophy and increased levels of triglycerides, glucose, and inflammatory markers [25,26]. 

What this implicates for metabolic imbalances and glycemic levels is also directly associated with the development of diabetes mellitus [27]. A longitudinal study with 369,362 children and adolescents in the United Kingdom found that patients with type 2 diabetes mellitus were mostly obese (47.1%) in comparison to individuals with age-appropriate BMI (4.33%), showing that since early life, overweight and obese individuals have a four-fold higher risk of developing diabetes mellitus. Moreover, the incidence of type 2 diabetes mellitus among overweight and obese people in this population increased from 6.4/100,000 in 1994–1998 to 33.2/100,000 people in 2009–2013 [28]. 

In addition to its impacts on this population’s quality of life, this increase also has an economic impact on health services. In 2018, the costs for the Brazilian Unified Health System (SUS) attributable to chronic non-communicable diseases reached BRL 3.45 billion, approximately USD 639 million in 2022. Of these costs, 59% were destined to the treatment of hypertension, 30% to diabetes, and 11% to obesity. Moreover, 72% of the costs related to those aged from 30 to 69 years and 56%, to women [29]. These data show a great medical care economic burden and the costs associated with reduced quality of life, disabilities, and social problems attributable to these diseases.

We emphasize limitations of our study. As in every cross-sectional study, it is impossible to establish a temporal sequence of events or analyze causality in the associated factors and effects of overweight and obesity. We performed no laboratory tests and measurements for diabetes and hypertension diagnoses. The study population reported this information based on previous medical diagnoses. The BMI was the only adiposity indicator used due to its ease and low cost of collection. However, measures such as the percentage of fat, skinfolds, and/or waist circumference could help to improve the accuracy of the adiposity of the interviewees. Despite these limitations, the literature lacks representative studies for the adult population in the western Amazon. The prevalence and associations discussed are consistent with other analyses and show great relevance for the control and prevention of overweight and obesity in the Amazon and other regions with similar socioeconomic and demographic characteristics.

## 5. Conclusions

The prevalence of overweight in adults in Rio Branco in 2019 is associated with factors such as high blood pressure, physical inactivity, age group, and not smoking. The factors associated with the prevalence of obesity were arterial hypertension and diabetes mellitus. Other factors make up the hierarchical model for obesity, such as female gender and smoking, but without statistical significance. Thus, this research can help in the control and prevention of overweight and obesity in the Amazon context or in other regions with similar socioeconomic and demographic characteristics.

## Figures and Tables

**Figure 1 nutrients-14-01079-f001:**
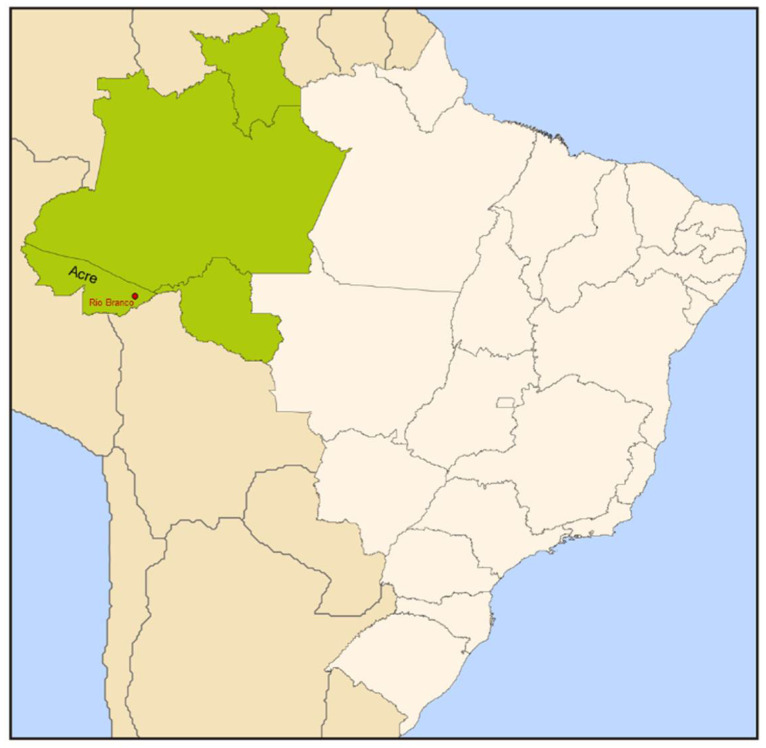
Map showing the location of Rio Branco, capital of the Acre State, in the western Brazilian Amazon (in green).

**Figure 2 nutrients-14-01079-f002:**
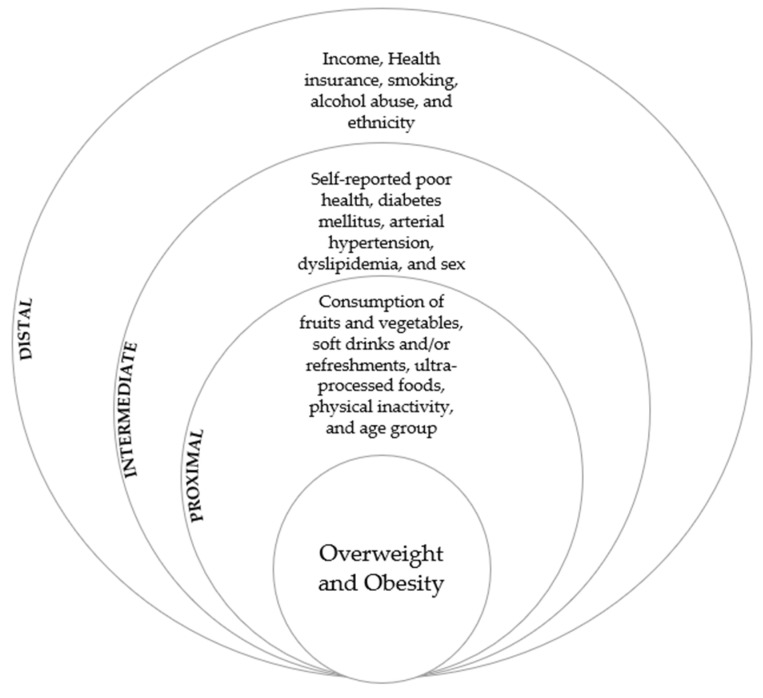
Hierarchical conceptual model for overweight and obesity.

**Table 1 nutrients-14-01079-t001:** Prevalence of overweight and obesity in adults in Rio Branco, Acre, 2019, according to sociodemographic and economic characteristics.

Variable	Total		Overweight				Obesity			
	n	N (%)	N (%)	Crude PR	95%CI	*p*	N (%)	Crude PR	95%CI	*p*
**Age group**										
18–24	158	52,896 (17.9)	18,960 (35.8)	1.00			5986 (11.3)	1.00		
25–39	385	91,920 (31.1)	51,983 (56.6)	1.58	1.21–2.05	<0.001	17,612 (19.2)	1.69	0.95–3.02	0.07
40–59	423	104,396 (35.3)	72,195 (69.2)	1.93	1.54–2.41	<0.001	28,717 (27.5)	2.43	1.41–4.19	<0.001
60+	251	46,324 (15.7)	28,866 (62.3)	1.74	1.34–2.26	<0.001	7029 (15.2)	1.34	0.78–2.30	0.28
**Sex**										
Male	538	139,014 (47.0)	82,726 (59.5)	1.00			23,330 (16.8)	1.00		
Female	679	156,523 (53.0)	89,279 (57.0)	0.96	0.83–1.10	0.54	36,013 (23.0)	1.37	1.04–1.81	0.03
**Ethnicity**										
Other	951	235,242 (79.6)	139,826 (53.4)	1.00			48,759 (20.7)	1.00		
White	266	60,295 (20.4)	32,178 (53.4)	0.9	0.77–1.05	0.17	10,585 (17.6)	0.84	0.64–1.12	0.24
**Per capita income**										
Up to ½ MW	306	81,868 (27.7)	46,696 (57.0)	1.00			14,765 (18.0)	1.00		
½–1 MW	318	78,830 (26.7)	43,679 (55.4)	0.97	0.79–1.20	0.78	17,109 (21.7)	1.20	0.85–1.71	0.30
1–2 MW	313	74,580 (25.2)	43,767 (58.7)	1.03	0.86–1.23	0.76	15,960 (21.4)	1.19	0.80–1.76	0.39
>2 MW	280	60,258 (20.4)	37,862 (62.8)	1.10	0.93–1.30	0.25	11,510 (19.1)	1.06	0.75–1.50	0.74
**Health insurance**										
No	988	243,533 (82.4)	139,770 (57.4)	1.00			49,411 (20.3)	1.00		
Yes	229	52,003 (17.6)	32,235 (62.0)	1.09	0.96–1.22	0.20	9933 (19.1)	0.94	0.70–1.26	0.69

n: unweighted count. N: expanded estimate value. (%): proportion from N. Crude PR: crude prevalence ratio. 95%CI: 95% confidence interval. *p*: Wald test value. MW: 2019 minimum wage (R$ 998.00).

**Table 2 nutrients-14-01079-t002:** Prevalence of overweight and obesity according to lifestyles and health of adults in Rio Branco, Acre, 2019.

Variable	Total		Overweight				Obesity			
	n	N (%)	N (%)	Crude PR	95%CI	*p*	N (%)	Crude PR	95%CI	*p*
**Regular fruit and vegetable consumption**										
No	1071	263,731 (89.2)	152,860 (58.0)	1.00			50,718 (19.2)	1.00		
Yes	146	31,806 (10.8)	19,144 (60.2)	1.04	0.88–1.22	0.64	8625 (27.1)	1.41	1.04–1.92	0.03
**Regular soft drink and/or refreshment consumption**										
No	1190	287,112 (97.1)	167,754 (58.4)	1.00			58,475 (20.4)	1.00		
Yes	27	8424 (2.9)	4251 (50.5)	0.86	0.55–1.36	0.52	869 (10.3)	0.51	0.13–1.95	0.32
**Regular ultra-processed food consumption**										
No	1076	257,754 (87.2)	152,391 (59.1)	1.00			53,549 (20.8)	1.00		
Yes	141	37,782 (12.8)	19,614 (51.9)	0.88	0.74–1.04	0.14	5794 (15.3)	0.74	0.51–1.07	0.11
**Smoking**										
No	1089	264,069 (89.4)	158,946 (60.2)	1.00			56,110 (21.2)	1.00		
Yes	128	31,468 (10.6)	13,059 (41.5)	0.69	0.49–0.97	0.03	3234 (10.3)	0.48	0.25–0.94	0.03
**Alcohol abuse**										
No	1034	249,601 (84.5)	146,858 (58.8)	1.00			49,472 (19.8)	1.00		
Yes	183	45,935 (15.5)	25147 (54.7)	0.93	0.78–1.10	0.40	9872 (21.5)	1.09	0.76–1.56	0.66
**Self-reported poor health**										
No	1137	279,637 (94.6)	163,459 (58.5)	1.00			56,000 (20.0)	1.00		
Yes	80	15,899 (5.4)	8545 (53.7)	0.92	0.67–1.25	0.59	3343 (21.0)	1.05	0.61–1.81	0.86
**Physical inactivity**										
No	838	204,845 (86.1)	113,265 (55.3)	1.00			40,127 (19.6)	1.00		
Yes	143	33,125 (13.9)	23098 (69.7)	1.26	1.08–1.47	<0.001	7533 (22.7)	1.16	0.80–1.68	0.42
**Diabetes mellitus**										
No	1015	242,751 (93.8)	144,168 (59.4)	1.00			48,954 (20.2)	1.00		
Yes	80	16,013 (6.2)	12,695 (79.3)	1.33	1.16–1.54	<0.001	6296 (39.3)	1.96	1.38–2.77	<0.001
**Arterial hypertension**										
No	906	228,793 (79.3)	120,326 (52.6)	1.00			38,537 (16.8)	1.00		
Yes	286	59,693 (20.7)	49,369 (82.7)	1.59	1.43–1.72	<0.001	20,475 (34.3)	2.04	1.61–2.56	<0.001
**Dyslipidemia**										
No	917	223,188 (88.3)	135,576 (60.7)	1.00			48,187 (21.6)	1.00		
Yes	154	29,463 (11.7)	21,251 (72.1)	1.19	1.03–1.37	0.01	7258 (24.6)	1.14	0.78–1.66	0.49

n: unweighted count. N: expanded estimate value. (%): proportion from N. Crude PR: crude prevalence ratio. 95%CI: 95% confidence interval. *p*: Wald test value.

**Table 3 nutrients-14-01079-t003:** Factors associated with overweight in adults from Rio Branco, Acre, 2019.

Variable	Distal Model 1		Intermediate Model 2		Proximal Model 2		Final Model	
	AdjPR	95%CI	AdjPR	95%CI	AdjPR	95%CI	AdjPR	95%CI
**Health insurance**								
Yes	1.00							
No	1.06	0.94–1.19						
**Ethnicity**								
Other	1.00							
White	0.88	0.76–1.02						
**Smoking**								
No	1.00		1.00		1.00		1.00	
Yes	0.69	0.49–0.97	0.71	0.55–0.92	0.62	0.41–0.93	0.62	0.41–0.93
**Diabetes mellitus**								
No			1.00					
Yes			1.09	0.95–1.23				
**Dyslipidemia**								
No			1.00					
Yes			1.04	0.91–1.19				
**Arterial hypertension**								
No			1.00		1.00		1.00	
Yes			1.46	1.34–1.58	1.45	1.31–1.62	1.45	1.31–1.61
**Regular ultra-processed food consumption**								
No					1.00			
Yes					0.93	0.75–1.15		
**Physical inactivity**								
No					1.00		1.00	
Yes					1.19	1.04–1.36	1.19	1.04–1.36
**Age group**								
18–24					1.00		1.00	
25–39					1.47	1.08–1.99	1.49	1.10–2.00
40–59					1.66	1.25–2.21	1.69	1.28–2.23
60+					1.34	0.97–1.86	1.37	1.01–1.87

AdjPR: adjusted prevalence ratio. 95%CI: 95% confidence interval.

**Table 4 nutrients-14-01079-t004:** Factors associated with obesity in adults from Rio Branco, Acre, 2019.

Variable	Distal Model 1		Intermediate Model 2		Proximal Model 2		Final Model	
	AdjPR	95%CI	AdjPR	95%CI	AdjPR	95%CI	AdjPR	95%CI
**Sex**								
Male	1.00		1.00		1.00		1.00	
Female	1.34	1.02–1.76	1.20	0.91–1.57	1.19	0.88–1.60	1.19	0.90–1.57
**Smoking**								
No	1.00		1.00		1.00		1.00	
Yes	0.50	0.26–0.96	0.74	0.41–1.34	0.59	0.28–1.23	0.69	0.37–1.28
**Dyslipidemia**								
No			1.00					
Yes			0.81	0.55–1.20				
**Diabetes mellitus**								
No			1.00		1.00		1.00	
Yes			1.61	1.15–2.22	1.61	1.11–2.33	1.52	1.08–2.13
**Arterial hypertension**								
No			1.00		1.00		1.00	
Yes			1.81	1.41–2.32	2.14	1.56–2.92	1.80	1.41–2.30
**Regular ultra-processed food consumption**								
No					1.00			
Yes					0.85	0.53–1.37		
**Physical inactivity**								
No					1.00			
Yes					1.07	0.76–1.49		
**Age group**								
18–24					1.00			
25–39					1.88	0.79–4.50		
40–59					2.07	0.85–5.03		
60+					0.88	0.32–2.36		

AdjPR: Adjusted prevalence ratio; 95%CI: 95% confidence interval.

## Data Availability

Data shown in this study are publicly and unrestrictedly available by the Brazilian Institute of Geography and Statistics (IBGE) on the website: https://www.ibge.gov.br/estatisticas/downloads-estatisticas.html?caminho=PNS/2019/Microdados/Dados (accessed on 6 February 2022).

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
