# Peer review of "Factors Associated with Overweight and Obesity in Adults from Rio Branco, Acre in the Western Brazilian Amazon"

_nutrients, 2022, doi:10.3390/nu14051079_

Round 1
Reviewer 1 Report
Dear Authors,
Thank you for your manuscript.
The paper is well-written. I have only a few minor comments.
The introduction is very scarce. Please add information explaining possible reasons for increasing rates of obesity and mechanisms explaining obesity-related comorbidities.
In the Methods section, the description of how study participants were selected and recruited is missing. Also, please explain whether the sample was representative of the region?
The description of the criteria on physical inactivity is incomplete. Please add information, whether the frequency, intensity, and duration of physical activity were taken into account when classifying respondents into the group physical inactivity "no"?
Next, in Table 3, together with the health insurance category "no", the upper limit of the CI is missing.
Author Response
Dear Reviewer,
We respectfully welcome and thank you for your valuable contributions, and for your suggestions and requests for corrections, which certainly qualify and give greater consistency to our manuscript.
The following describes a point-by-point response to the reviewer's comments and the changes made.
Point 1: The introduction is very scarce. Please add information explaining possible reasons for increasing rates of mechanisms and explaining obesity-related comorbidities.
Response 1: The paragraph has been rewritten: “In Brazil, according to the Surveillance System of Risk and Protective Factors for Chronic Diseases by Telephone Survey (Vigitel), the prevalence of obesity in individuals aged 18 years or older increased from 11.8% in 2006 to 20.3% in 2019, a 72% increase. Moreover, the number of people with obesity has grown in all regions of the country. The trend that may be associated with urbanization, increased consumption of ultra-processed foods, reduced consumption of natural foods and increased physical inactivity. A nutritional transition that involves behavioral, geographic, cultural and socioeconomic factors. Thus, regional iferences must be considered and analyzed individually [6-7].”
Point 2: In the Methods section, the description of how study participants were selected and recruited is missing. Also, please explain whether the sample was representative of the region?
Response 2: The paragraph has been rewritten: “The PNS is a household survey which employs a cluster sampling plan in three selection stages: I) primary sampling units (census tracts – UPA), II secondary units (households/UPA), and III) tertiary units (resident aged 15 years or older who answered individual questionnaires). Thus, the information collected included questions on the household, its residents, and the adult drawn. Initially, contact was made with a resident of the selected household, and the study and its objectives were presented. With the consent to participate, the household questionnaire was applied to the informant and the individual questionnaire to the selected resident. For this study, respondents aged 18 years or older were selected, excluding pregnant women and women with suspected pregnancy. As it is a complex sample, estimates were considered by the weight on eligible adult residents, corresponding to the capital and state analyzed, enabling a representative sample of the population of Rio Branco, Acre, in 2019 [10].”
Point 3: The description of the criteria on physical inactivity is incomplete. Please add information, whether the frequency, intensity, and duration of physical activity were taken into account when classifying respondents into the group physical inactivity "no"?
Response 3: The paragraph has been rewritten: “Physical inactivity was assessed through questions related to frequency (days per week) and duration (time per day) in different domains: such as free time or leisure, occupational activity, during commuting and within the scope of domestic activities. This dichotomous variable was considered for the individual who practiced physical activity for less than 20 minutes a day, or who did not practice any activity in the last three months prior.”
Point 4: Next, in Table 3, together with the health insurance category "no", the upper limit of the CI is missing.
Response 4: The information was added in table 3, where PRadj1.06; 95% CI 0.94-1.19.
Sincerely,
The authors.
Reviewer 2 Report
This is a very well conceptualized and executed study of factors associated with overweight and obesity in a specific population. The data is presented in a clear and concise manner and the authors are commended for the concept, statistical analyses and clarity of presentation.
I only have a few comments to suggest that in my opinion would strengthen the manuscript.
- The use of BMI as an indicator of adiposity has been under scrutiny in the recent past. Perhaps the authors could discuss as part of the limitations section that other indices of adiposity may be more reflective of overweight and obesity (i.e. waist to hip circumference etc.).
- The categorization of all non-white individuals into "other" may warrants a bit more clarification/justification. The same with is with the binary categorization for sex. Could the authors perhaps discuss these two points, or at the very least include a rationale for such design.
Author Response
Dear Reviewer,
We respectfully welcome and thank you for your valuable contributions, and for your suggestions and requests for corrections, which certainly qualify and give greater consistency to our manuscript.
The following describes a point-by-point response to the reviewer's comments and the changes made.
Point 1:The use of BMI as an indicator of adiposity has been under scrutiny in the recent past. Perhaps the authors could discuss as part of the limitations section that other indices of adiposity may be more reflective of overweight and obesity (i.e. waist to hip circumference etc.).
Response 1: The limitation on the exclusive use of this indicator was detailed in the manuscript: “The BMI was the only adiposity indicator used, due to its ease and low cost of collection. However, measures such as the percentage of fat, skinfolds and/or waist circumference could help to improve the accuracy of the respondents' adiposity.”
Point 2: The categorization of all non-white individuals into "other" may warrants a bit more clarification/justification. The same with is with the binary categorization for sex. Could the authors perhaps discuss these two points, or at the very least include a rationale for such design.
Response 2: The paragraph has been rewritten: “Gender was collected using the instrument's binary alternatives: male or female, since the database used does not work with gender; skin color was collected by the alternatives: white, yellow, brown and indigenous, and for analysis-analysis the white category and the others were recategorized to non-white. In view of the difficulties of self-perception and phenotypic self-declaration of skin color, especially in multiracial countries such as Brazil [11-12].”
Sincerely,
The authors.